# Linearizing Contextual Bandits with Latent State Dynamics

**Elliot Nelson**[1]     **Debarun Bhattacharjya**[1]     **Tian Gao**[1]     **Miao Liu**[1]     **Djallel Bouneffouf**[1]     **Pascal Poupart**[2]

[1]IBM T. J. Watson Research Center, Yorktown Heights, NY, USA
[2]David R. Cheriton School of Computer Science, University of Waterloo, Waterloo, ON, Canada

## Abstract

In many real-world applications of multi-armed bandit problems, both rewards and contexts are often influenced by confounding latent variables which evolve stochastically over time. While the observed contexts and rewards are nonlinearly related, we show that prior knowledge of latent causal structure can be used to reduce the problem to the linear bandit setting. We develop two algorithms, Latent Linear Thompson Sampling (L$^2$TS) and Latent Linear UCB (L$^2$UCB), which use online EM algorithms for hidden Markov models to learn the latent transition model and maintain a posterior belief over the latent state, and then use the resulting posteriors as context features in a linear bandit problem. We upper bound the error in reward estimation in the presence of a dynamical latent state, and derive a novel problem-dependent regret bound for linear Thompson sampling with non-stationarity and unconstrained reward distributions, which we apply to L$^2$TS under certain conditions. Finally, we demonstrate the superiority of our algorithms over related bandit algorithms through experiments.

## 1 INTRODUCTION

Multi-armed bandits have been successfully applied in domains such as healthcare [Durand et al., 2018, Zhu et al., 2018], finance [Shen et al., 2015], and recommender systems [Zhou et al., 2017]. In this work, we are interested in contextual multi-armed bandit problems where the presence of a latent variable is crucial for predicting rewards. Furthermore, it is typical in many real-world problems for additional complexity to arise in the form of latent variable non-stationarity (dynamics). Consider the following illustrative real-world applications:

- An interactive AI agent for personalized education chooses material to help a student's evolving state of knowledge, using observations such as the time taken to answer questions.
- A rover on a mission explores blocks of land, taking samples for information about the ore grade and choosing real-time mining strategies for each block.
- A recommender system selects items for users with evolving latent preferences or values, potentially using observable signals such as behavior patterns.

Such problems can be represented with the graphical model of Figure 1. Here a decision-making agent must use additional side information or context data (denoted $x$) for inference of an unseen, time-dependent latent state (denoted $z$), in order to improve reward predictions.

Our approach to the non-stationary latent bandit problem of Figure 1 focuses on leveraging prior knowledge of the graphical structure to apply simpler methods to a difficult problem, using a strategy of reduction to a known problem. The linear multi-armed bandit setting [Auer, 2002, Abbasi-Yadkori et al., 2011] has been studied extensively, leading to many algorithms and related theoretical guarantees. While complex real-world tasks generally involve nonlinear relationships between observed variables and target objectives (such as the nonlinear relationship between $x_t$ and $r_t$ in Figure 1), a key motivating observation for our work is that expected values, and in particular expected rewards, are linearly related to probabilities of unknown variables or parameters. This linear relationship can be exploited using algorithms and theoretical analyses for the linear bandit setting.

For the non-stationary bandit task of Figure 1, this requires maintaining posterior probabilities over the current latent state $z_t$. Since Figure 1 may be viewed as an extension of a hidden Markov model (HMM) [Rabiner, 1989] into a multi-armed bandit task, we leverage existing methods for online learning of HMMs. In particular, online expectation maximization (EM) is an established method which learns

*Accepted for the 38$^{th}$ Conference on Uncertainty in Artificial Intelligence* (UAI 2022).

to perform approximate Bayesian inference over the latent state, when applied in our setting.

**Contributions.** We combine existing methods for hidden Markov models and linear bandit problems in a novel way, to make the following contributions: (i) We identify conditions under which contextual multi-armed bandit problems with an evolving hidden state (Figure 1) can be mapped to a linear bandit problem. (ii) We introduce novel bandit algorithms for the setting of Figure 1, Latent Linear Thompson Sampling ($L^2$TS) and Latent Linear Upper Confidence Bounds ($L^2$UCB), which combine approximate online Bayesian inference over the latent state with linear bandit methods, and demonstrate superior performance compared to baseline algorithms. (iii) We derive a high-probability bound (Theorem 1) on least-squares parameter estimation error in the setting of Figure 1. (iv) We derive a novel problem-dependent regret bound for linear Thompson sampling with non-stationary and arbitrary reward distributions, and apply it to $L^2$TS (Theorem 2).

In the next section, we discuss the advantages of our approach and limitations of existing multi-armed bandit approaches in settings where a time-evolving latent state influences contexts and rewards.

## 2   RELATED WORK

*Linear Bandits.* Our work identifies a path for applying methods and analysis for the linear bandit framework [Auer, 2002, Abbasi-Yadkori et al., 2011] to a larger class of (nonlinear) contextual bandit problems. We introduce algorithms which use the linear Thompson sampling algorithm of Agrawal and Goyal [2013b] or the related LinUCB algorithm [Li et al., 2010, Chu et al., 2011] as subroutines. While linear bandit methods have been applied in various settings, our approach of leveraging linearity with respect to posterior probabilities is novel, as well as application of the suite of linear bandit tools to latent bandit problems.

*Non-Stationary Bandits.* The decision-making problem of Figure 1 lies at the intersection of the (more general) class of contextual bandit problems, in which additional context information is available along with reward data, and the class of *non-stationary bandit* [Auer et al., 2003, Luo et al., 2018, Hartland et al., 2007, Garivier and Moulines, 2008, Yu and Mannor, 2009] problems, which introduce time-dependence into the reward distribution. The bulk of existing work in non-stationary bandits focuses on detecting change in distributions or parameters [Luo et al., 2018]. In our setting, these methods are limited, as they cannot model the latent causal structure, which allows for improved modeling and prediction of distributional change over time.

*Latent Bandits.* A growing body of research on *latent bandit* [Maillard and Mannor, 2014, Zhou and Brunskill, 2016]

problems seeks to model reward distributions which are influenced by a latent state, as in Figure 1. Most work in this area does not consider the case of dynamical state transitions. The graphical structure of Figure 1 is considered in Hong et al. [2021], which in contrast to the present work, focuses on off-policy learning. Other recent work [Hong et al., 2020] (see also Hong et al. [2020]) considers a closely related problem in which a dynamical hidden state influences rewards, but assuming a different graphical structure in which contexts are unaffected by the latent state (and thus cannot be leveraged for inference of $z$). Our approach is similar to that of Hong et al. [2020], in that we use Thompson sampling [Thompson, 1933, Chapelle and Li, 2011, Russo and Roy, 2014] as an exploration heuristic. However, their approach involves Thompson sampling of latent states as well as parameters. In settings where latent states changes occur frequently, such exploration of the latent space may not yield significant information gain before the state changes again,[1] and can thus under-exploit. Furthermore, the practical algorithm proposed in Hong et al. [2020] uses particle filtering [Doucet et al., 2001], which can struggle to scale to higher dimensions with a fixed number of particles. In comparison, we sidestep the difficulties of approximating a high-dimensional posterior by selectively maintaining uncertainty over the most task-relevant unknowns. Moreover, in the asymptotic limit of long sequences, the cumulative log-likelihood $L(\mu_\star) = \sum_t \log p(r_t|a_t; \mu_\star)$ of reward data becomes large, causing the posterior $p(\mu_\star) \propto e^{L(\mu_\star)}$ over reward parameters $\mu_\star$ to satisfy the Laplace approximation and generally converge to a Gaussian form. We exploit this asymptotic property with linear Thompson sampling [Agrawal and Goyal, 2013a], which uses a multivariate normal posterior.

*Recommender Systems.* The graphical structure of our problem, with a latent variable acting as a confounder of context observations and rewards, is shared in the literature on bandit algorithms for recommender systems (e.g. [Sen et al., 2017, Kawale et al., 2015]). In comparison to these works, which generally assume i.i.d. latent variables, our work is primarily an extension in the direction of non-stationarity.

*Causal Bandits.* Lastly, our work is also related to the burgeoning area of *causal bandits* [Lattimore et al., 2016] where causal mechanisms are explicitly modeled. Confounding from a latent variable was considered in Bareinboim et al. [2015], Lee and Bareinboim [2018], Sen et al. [2017], but under the assumption of i.i.d. data (no non-stationarity), and in an offline rather than online learning setting.

*Theoretical Analysis.* Discussion of related work on the subject of regret bounds is deferred to Section 5.2.

---

[1]In contrast, information gained via exploration about fixed parameters will not become outdated.

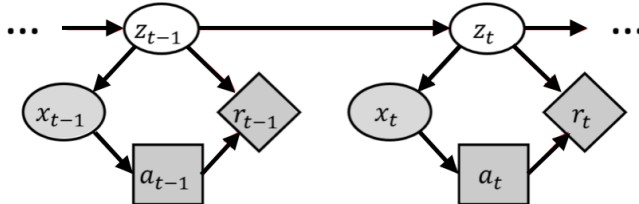

Figure 1: An influence diagram representation of the non-stationary version of our latent bandit setting. The latent state $z$ changes dynamically while context $x$ is observed at the time of choosing action $a$ (rectangle), represented by the informational arc from $x$ to $a$. Reward $r$ (diamond) is a function of $a$ and $z$. Shaded (white) nodes indicate observed (unobserved) variables.

## 3   PROBLEM SETTING

In this section, we describe our contextual multi-armed bandit problem setting with a dynamical latent state (Section 3.1), describe a related linear bandit problem setting (Section 3.2), and show that the latent bandit setting of Section 3.1 can be reduced to the linear bandit setting of Section 3.2 under certain conditions (Section 3.3).

### 3.1   NON-STATIONARY LATENT BANDITS

We consider the non-stationary bandit environment of Figure 1 in which a dynamical latent state $z$ acts as a confounder of observations (or contexts) $x$ and rewards $r$. The figure is represented as an influence diagram, which is a graphical model for decision making under uncertainty [Howard and Matheson, 1984]. At any epoch, context $x$ is observed before selecting action $a$, and reward $r$ depends on $a$ and $z$.

While the context and reward may be either discrete or real-valued[2], the latent state $z \in \mathcal{Z} = \{1, ..., Z\}$ and action $a \in \mathcal{A} = \{1, ..., K\}$ are assumed to be discrete. The latent state $z$ evolves stochastically according to a transition matrix $\Phi^\star$ (assumed to be ergodic) with elements, $p(z_t = z' | z_{t-1} = z; \phi^\star) = \phi_{z,z'}^{(\phi)\star}$. The equilibrium distribution $\rho_{\text{eq}}^{(\phi)}(z)$ for a given matrix $\Phi$ is the stationary distribution, $\Phi\rho_{\text{eq}}^{(\phi)} = \rho_{\text{eq}}^{(\phi)}$. (For any categorical distribution $p(z)$, we will denote by $p \in \mathbb{R}^Z$ the vector whose elements are the probabilities $p(z)$.) Given $z$, an observed context $x$ is generated from a conditional distribution $p(x|z; \theta^\star)$ with parameters $\theta^\star$. Lastly, rewards are generated from conditional distributions $p(r|z, a)$; we denote their expected values as $(\mu_\star^{(a)})_z := \mathbb{E}[r|z, a]$, with $\mu_\star^{(a)} \in \mathbb{R}^Z$ being an action-wise vector of means, and variance as $\text{Var}[r|z, a]$. We collectively denote the action-wise parameter vectors as $\mu_\star := \{\mu_\star^{(a)}\}_{a=1}^K$.

Our algorithm relies on the estimation and use of a posterior belief, $p_t(z|x_{1:t}) := p(z_t = z|x_{1:t})$ over the current latent state, which is a categorical distribution represented as a $Z$-dimensional vector. Given a transition model $p(z'|z; \hat{\phi})$ and observation model $p(x|z; \hat{\theta})$, it can be updated every timestep with Bayes' rule:[3]

$$\hat{p}_t(z|x_{1:t}) \propto \sum_{z'} \hat{p}_{t-1}(z'|x_{1:t-1})\hat{\phi}_{z,z'}p(x_t|z; \hat{\theta}) \quad (1)$$

where the hat notation denotes model estimates. We will distinguish the model posterior $\hat{p}$ from the "true" posterior

$$p_t^\star(z) := p(z_t = z|x_{1:t}; \phi^\star, \theta^\star, \rho_0), \quad (2)$$

which uses ground truth parameters and the true prior, $p_0^\star(z) := \rho_0(z)$.

A policy $\pi$ is a mapping from partial histories $(x_{1:t}, r_{1:t-1}, a_{1:t-1})$ at any time $t$ to probabilities of selecting each action, $a_t = a$. The optimal policy $\pi^\star$ is defined as the policy which selects, at every timestep, the action with highest expected reward, given the true parameters (but without accessing the true latent state), that is, $a_t^\star := \text{argmax}_a (p_t^\star)^\top \mu_\star^{(a)}$. We will quantify performance with expected cumulative regret, defined – for any policy $\pi$ – as the loss in expected rewards after $T$ timesteps relative to the optimal policy: $\mathcal{R}_\pi(T) := \sum_{t \leq T} (\mathbb{E}_{\pi^\star}[r_t] - \mathbb{E}_\pi[r_t])$.

### 3.2   LINEAR BANDITS

We will apply methods from the linear bandit setting to the contextual latent bandit setting of Section 3.1, in which observations $x_t$ and reward $r_t$ are nonlinearly related. We work with a slightly modified linear bandit setting as compared to the typical setting in the literature [Agrawal and Goyal, 2013b]: At each timestep, a context feature vector $c_t \in \mathbb{R}^d$ is observed, an action $a_t = a$ is selected from $K$ possible actions, and a reward

$$r_t = c_t^\top \mu_\star^{(a)} + \epsilon_t \quad (3)$$

with mean value $c_t^\top \mu_\star^{(a)}$ is observed. The random noise vector $\epsilon_t \in \mathbb{R}^d$ has mean zero by definition, $\mathbb{E}[\epsilon_t] = 0$, but need not satisfy any other conditions such as sub-Gaussianity or i.i.d. data across time. In order to maximize returns, the agent must use the sequential context data $c_{1:t}$ to learn the unknown mean reward parameters $\mu_\star^{(a)} \in \mathbb{R}^d$ for each action $a$.[4] Given the context $c_t$, the corresponding optimal action is $a_t^\star := a^\star(c_t) := \text{argmax}_a c_t^\top \mu_\star^{(a)}$.

---

[2]We denote context as a scalar for simplicity but our work is equally applicable to settings with high-dimensional observations.

[3]We occasionally use $\propto$ to denote equality up to a normalizing constant.

[4]In other variations of the linear bandit setting, the same parameters $\mu$ may be shared across actions, while a separate per-action context $c_t^{(a)}$ may be observed.

In Section 4, we introduce algorithms which use linear Thompson sampling (LinTS) [Agrawal and Goyal, 2013b] or LinUCB [Li et al., 2010, Chu et al., 2011] as subroutines. LinUCB and LinTS use observed contexts and rewards to maintain (for each action) a least-squares estimator:

$$\hat{\mu}^{(a)} = (B^{(a)})^{-1} f^{(a)}, \qquad (4)$$

where $f^{(a)} := \sum_{t'=1}^{t} \mathbf{1}(a_{t'} = a) c_{t'} r_{t'}$, with $\mathbf{1}(A)$ being the indicator function equal to 1 (0) when $A$ is true (false), and $B^{(a)} := \lambda_\mu \mathbb{1}_d + \sum_{t'=1}^{t} \mathbf{1}(a_{t'} = a) c_{t'} c_{t'}^\top$ is an empirical co-variance matrix (we assume $\lambda_\mu > 0$ to ensure invertibility). LinUCB uses the estimator covariance to compute upper confidence bounds, while LinTS uses each estimator $\hat{\mu}^{(a)}$ to Thompson sample from a multivariate Gaussian posterior, $\mu^{(a)} \sim \mathcal{N}(\hat{\mu}^{(a)}, (B^{(a)})^{-1})$, and selects at each timestep the corresponding optimal action: $a_t = \text{argmax}_a c_t^\top \mu^{(a)}$.

## 3.3 REDUCTION TO THE LINEAR BANDIT PROBLEM

We now exploit the linear relationship between rewards and probabilities over the latent space to show that the latent bandit problem of Section 3.1 can be reduced to the linear bandit setting of Section 3.2.

**Lemma 1.** *When the true model parameters $(\theta^\star, \phi^\star)$ and initial latent state probabilities $\rho_0(z) = p(z_0 = z)$ in the model from Figure 1 are known, the latent bandit setting of Section 3.1 reduces to the linear bandit setting of Section 3.2.*

*Proof.* Conditional on a sequence of observations $x_{1:t}$ in the latent bandit setting and action $a_t = a$, the reward $r_t$ is generated from the mixture distribution

$$p(r_t = r | a_t = a, x_{1:t}; \theta^\star, \phi^\star) = \sum_z (c_t)_z p(r|z, a),$$

where we have defined $c_t \in \mathbb{R}^Z$ as the vector with elements equal to the posterior probabilities

$$(c_t)_z := p(z_t = z | x_{1:t}; \theta^\star, \phi^\star) := p_t^\star(z). \qquad (5)$$

The expected reward at time $t$ is therefore

$$\mathbb{E}[r_t | a_t = a, x_{1:t}; \theta^\star, \phi^\star] = \sum_z (c_t)_z (\mu_\star^{(a)})_z = c_t^\top \mu_\star^{(a)}.$$

Thus, the reward takes the form of Eq. (3), with $d = Z$ being the number of latent states, $c_t$ defined in Eq. (5), and $\mu_\star^{(a)} \in \mathbb{R}^Z$ being the vector of latent-conditioned mean rewards $(\mu_\star^{(a)})_z$. $\qquad \square$

Lemma 1 shows that the posterior belief over the current latent state $z_t$ can be viewed as a compression of the context history $x_{1:t}$ into a (nonlinearly) transformed context variable

which is related linearly to rewards. Since Lemma 1 assumes access to the true parameters $(\theta^\star, \phi^\star)$, in general it will only apply in the asymptotic limit ($t \to \infty$) in which $(\theta^\star, \phi^\star)$ have been learned. Prior to this asymptotic regime, error in model estimates of these parameters will corrupt the context features $c_t$ in the corresponding linear bandit problem with noise and/or systematic bias.

We end this section by noting that the space of context vectors $c_t$, or equivalently posterior beliefs $p_t^\star$ (see Eq. (5)), is partitioned into subspaces – denoted $\mathcal{P}_{a^\star}$ – for which action $a^\star$ is optimal, i.e. $a^\star = \text{argmax}_a c_t^\top \mu_\star^{(a)}$. In the following section, we will build on Lemma 1 to develop a latent bandit algorithm which estimates rewards, Eq. (4), with contexts $c_t \to p_t^\star$ as in Eq. (5).

# 4 LATENT LINEAR BANDIT ALGORITHMS

Since the non-stationary latent bandit problem of Section 3.1 can be reduced to the linear bandit setting as long as an accurate posterior belief over the latent state $z$ can be maintained, algorithms for the latent bandit problem can be built by combining (i) methods for approximate inference over $z$ with (ii) linear bandit algorithms. In this paper, we introduce two specific such algorithms, which use (i) Online Expectation Maximization (EM) for learning the parameters $(\theta^\star, \phi^\star)$ of a hidden Markov model (and thus learning the "true" posteriors $p_t^\star(z)$ assumed in Lemma 1), and (ii) either LinTS or LinUCB, into an end-to-end pipeline.

*Latent State Inference.* We use the online EM algorithm of Mongillo and Deneve [2008] (for categorical context data), and the related Algorithm 1 of Cappé [2011] (for continuous context data). As indicated in Algorithms 1 and 2, after observing $x_t$ these online EM algorithms recursively update (i) the vector estimate $\hat{p}_t$ of latent state probabilities, (ii) sufficient statistics $\hat{\psi}_t$, and (iii) parameter estimates $(\hat{\theta}, \hat{\phi})$ (determined by $\hat{\psi}_t$). Further details, including the form of sufficient statistics $\hat{\psi}_t$ for multinomial or Gaussian distributions, are provided in Appendix A. Importantly, the approximate Bayes' update of the model posterior over the latent state, Eq. (1), takes place as part of the online EM update. After observing the reward $r_t$, the model posterior $\hat{p}_t$ is again updated using a reward likelihood model $p(r|z, a; \hat{\mu})$ which is either Bernoulli or Gaussian in our experiments (see Appendix B).

*Thompson Sampling and UCB.* As described in Section 3.3, we use the model posterior over the current latent state $\hat{p}_t$ as a context feature vector in the linear bandit setting, $c_t = \hat{p}_t$, and apply either linear Thompson Sampling [Agrawal and Goyal, 2013b] (L[2]TS, Algorithm 1) or LinUCB [Li et al., 2010, Chu et al., 2011] (L[2]UCB, Algorithm 2) as exploration heuristics to select actions. Like L[2]TS, L[2]UCB treats the posterior beliefs $\hat{p}_t$ as context vectors in a linear bandit

**Algorithm 1:** Latent Linear Thompson Sampling (L$^2$TS)

**Input:**
  Prior over latent state, $\hat{p}_0 \in [0,1]^Z$
  Initial parameter estimates $(\hat{\theta}, \hat{\phi})$
  Initial sufficient statistics $\hat{\psi}_0$
  $f^{(a)} = \mathbf{0}_Z$, $B^{(a)} = \lambda_\mu \mathbb{1}_Z$, for $a \in \mathcal{A}$; $\lambda_\mu > 0$
  Likelihood variance $\tilde{\sigma}_r > 0$
**for** $t \leftarrow 1, 2, ...$ **do**
  Observe $x_t$;
  Update posterior $\hat{p}_t$ and parameters $(\hat{\theta}, \hat{\phi})$:
    $(\hat{\theta}, \hat{\phi}, \hat{p}_t, \hat{\psi}_t) = \text{OnlineEM}(x_t; \hat{\theta}, \hat{\phi}, \hat{p}_{t-1}, \hat{\psi}_{t-1})$
  Sample $\mu^{(a)} \sim \mathcal{N}(\hat{\mu}^{(a)}, \tilde{\sigma}_r^2 (B^{(a)})^{-1})$ for $a \in \mathcal{A}$
  Select action $a = \text{argmax}_{a'} \hat{p}_t^\top \mu^{(a')}$
  Observe $r_t$
  Update mean reward estimates:
    $B^{(a)} \leftarrow B^{(a)} + \hat{p}_t \hat{p}_t^\top$,    $f^{(a)} \leftarrow f^{(a)} + \hat{p}_t r_t$
    $\hat{\mu}^{(a)} = (B^{(a)})^{-1} f^{(a)}$
  Update posterior, $\hat{p}_t(z) \propto \sum_{z'} \hat{p}_t(z') p(r|z, a; \hat{\mu})$

**Algorithm 2:** Latent Linear UCB (L$^2$UCB)

**Input:**
  Prior over latent state, $\hat{p}_0 \in [0,1]^Z$
  Initial parameter estimates $(\hat{\theta}, \hat{\phi})$
  Initial sufficient statistics $\hat{\psi}_0$
  $f^{(a)} = \mathbf{0}_Z$, $B^{(a)} = \lambda_\mu \mathbb{1}_Z$, for $a \in \mathcal{A}$; $\lambda_\mu > 0$
  Exploration parameter $\alpha_{\text{UCB}} > 0$
**for** $t \leftarrow 1, 2, ...$ **do**
  Observe $x_t$;
  Update posterior $\hat{p}_t$ and parameters $(\hat{\theta}, \hat{\phi})$:
    $(\hat{\theta}, \hat{\phi}, \hat{p}_t, \hat{\psi}_t) = \text{OnlineEM}(x_t; \hat{\theta}, \hat{\phi}, \hat{p}_{t-1}, \hat{\psi}_{t-1})$
  Compute upper confidence bounds,
    $\pi_a = \hat{p}_t^\top \hat{\mu}^{(a)} + \alpha_{\text{UCB}} \sqrt{\hat{p}_t^\top (B^{(a)})^{-1} \hat{p}_t}$
  Select action $a = \text{argmax}_{a'} \pi_{a'}$
  Observe $r_t$
  Update reward estimator & covariance:
    $B^{(a)} \leftarrow B^{(a)} + \hat{p}_t \hat{p}_t^\top$,    $f^{(a)} \leftarrow f^{(a)} + \hat{p}_t r_t$
    $\hat{\mu}^{(a)} = (B^{(a)})^{-1} f^{(a)}$
  Update posterior, $\hat{p}_t(z) \propto \sum_{z'} \hat{p}_t(z') p(r|z, a; \hat{\mu})$

---

problem, and uses the same reward estimators $\{\hat{\mu}^{(a)}\}$ and covariance matrices $B^{(a)}$. The differences between L$^2$TS and L$^2$UCB are highlighted in blue in Algorithms 1 and 2. Note that L$^2$UCB asymptotically selects the action with the highest expected reward $\hat{p}_t^\top \hat{\mu}^{(a)} = \sum_z \hat{p}_t(z) \hat{\mu}_z^{(a)}$ given the current posterior vector $\hat{p}_t$, and assigns an exploration bonus to actions whose reward estimates $\hat{\mu}_z^{(a)}$ it is less certain of (in terms of the covariance $(B^{(a)})^{-1}$), for states $z$ that have high probability $\hat{p}_t(z)$.

We emphasize that while online EM only maintains point estimates $(\hat{\theta}, \hat{\phi})$, L$^2$TS and L$^2$UCB use exploration heuristics which leverage uncertainty in reward parameters $\{\hat{\mu}^{(a)}\}$ and in the current latent state $z_t$. In comparison, the algorithm of Hong et al. [2020] also maintains Bayesian uncertainty over the transition matrix, requiring a more computationally intensive particle filtering implementation. Our more computationally lightweight approach focuses on maintaining task-relevant uncertainty over $(z_t; \mu_\star)$ (see Section 2), and performed best empirically (Section 6). The computational complexity of L$^2$TS and L$^2$UCB is polynomial in the number of latent states $Z$ (due to the online EM updates shown in Appendix A; see Cappé [2011] for further discussion) and independent of the time $t$, making these algorithms scale well in problems with very long time horizons and low-dimensional latent structure.

## 5 THEORETICAL ANALYSIS

In this section, we (i) demonstrate that linear bandit reward estimation can be effectively applied to the non-stationary latent bandit setting from Figure 1 by upper bounding the error of the reward estimators used by L$^2$TS and L$^2$UCB

(Theorem 1), and (ii) derive a high-probability regret bound for linear Thompson sampling, using Theorem 1 to apply it to L$^2$TS.

### 5.1 REWARD ESTIMATION ERROR

In the case of a dynamical latent state, the reduction to the linear bandit setting described in Section 3.3 results in contexts $c_{1:t}$ and rewards $r_{1:t}$ which are not i.i.d. across time. Here, we state a result which shows that reward estimation via reduction to the linear bandit setting will converge to the true reward parameters $\{\mu_\star^{(a)}\}$ given a sufficiently long time horizon:

**Theorem 1.** *Assuming that (i) the latent state Markov chain is ergodic and in equilibrium, $z_1 \sim \rho_{\text{eq}}^{(\phi)}(\cdot)$, and when (ii) the true parameters $(\theta^\star, \phi^\star)$ are known and are used to compute $\hat{\mu}^{(a)}$ [Eq. (4) with $c_t \to p_t^\star$, in Eq. (5)], the error in $\hat{\mu}^{(a)}$ at time $t = T$ for any algorithm which selects the optimal action given $x_{1:T}$ with probability at least $\pi_{\min}$, is upper bounded,[5]*

$$|\hat{\mu}_z^{(a)} - (\mu_\star^{(a)})_z| \quad (6)$$
$$< \frac{2Z^2}{\pi_{\min}^2 \lambda_{\min}^{(a)}} \sqrt{\frac{1}{\delta \cdot T} \left( \sigma_{\text{eq}}^2 + ||\mu_\star^{(a)}||_1^2 \frac{4}{\gamma_{\phi^\star}} \left( 1 + \log \zeta_{\phi^\star} \right) \right)}$$

*for any $z$ with probability at least*

$$1 - \delta - \frac{8Z^3}{\pi_{\min}^2 \lambda_{\min}^{(a)}} \frac{1}{T \gamma_{\phi^\star}} (\kappa + \log \log(1/\rho_{\min})) \quad (7)$$

*for any $\delta \in (0, 1)$. Here, $\kappa \approx 6.8$, $\zeta_{\phi^\star}$ is a $\Phi^\star$-dependent numerical constant (see Appendix C.2), $\rho_{\min} := \min_z \rho_{\text{eq}}^{(\phi)}(z)$*

---

[5]Here, $||\mu||_\ell$ denotes the $\ell$-norm of a vector $\mu$.

is the equilibrium probability of the least probable latent state, $\sigma_{\text{eq}}^2 := \max_a \sum_z \rho_{\text{eq}}^{(\phi)}(z)\text{Var}[r|z,a]$ is a measure of reward noise when the latent state is in equilibrium, $\lambda_{\min}^{(a)} = \lambda_{\min}^{(a)}(T)$ is the minimal eigenvalue of the action-wise asymptotic expected inverse covariance matrix[6]

$$\bar{B}^{(a)}(T) := \frac{1}{T}\sum_{t=1}^{T}\mathbb{E}_{x_{1:t}\sim\rho_{\text{eq}}^{(\phi)}}[\mathbf{1}(p_t^\star \in \mathcal{P}_a)p_t^\star(p_t^\star)^\top], \quad (8)$$

*averaged over histories generated from the equilibrium distribution, and $\gamma_\phi := \min_{z_1,z_2}\sum_z \min(\phi_{z,z_1},\phi_{z,z_2})$ is the minimal mixing rate of a transition matrix $\Phi$ [Boyen and Koller, 1998].*

*Proof (Outline).* Appendix C has the complete proof. The derivation relies primarily on a KL divergence contraction theorem for stochastic Markov processes from Boyen and Koller [1998] to show that posterior probabilities used to compute the estimators $\hat{\mu}^{(a)}$ are approximately uncorrelated, $\mathbb{E}[p^\star(z)p_{t'}^\star(z')] \approx \mathbb{E}[p_t^\star(z)]\mathbb{E}[p_{t'}^\star(z')]$, over time separations $|t-t'|$ greater than the minimal mixing time $1/\gamma_{\phi^\star}$. Thus, the quantities $f^{(a)}$ and $B^{(a)}$ in Eq. (4) are sums of approximately independent random variables over blocks of at least $1/\gamma_{\phi^\star}$ timesteps. We quantify this with upper bounds on the variances $\text{Var}[f^{(a)}]$ and $\text{Var}[B^{(a)}]$ across context and reward histories, apply Chebyshev's inequality to obtain high-probability bounds on the deviation of $f^{(a)}$ and $B^{(a)}$ from their expected values at large $T$, and derive an eigenvalue bound for the inverse matrix $(B^{(a)})^{-1}$ in order to upper bound the deviation of the product $\hat{\mu}^{(a)} = (B^{(a)})^{-1}f^{(a)}$ from $\mu_\star^{(a)}$. $\quad\quad\square$

Theorem 1 describes the effect of the latent dynamics and resulting posterior beliefs $p_t^\star$ on reward parameter estimation. At times $T$ sufficiently large compared to the mixing time $t_{\phi^\star} := 1/\gamma_{\phi^\star}$, correlations between posterior beliefs (i.e. the dependent variables in linear regression estimation of $\mu_\star^{(a)}$) at different times are small, and reward data are close to i.i.d., allowing for a $1/\sqrt{T}$ error reduction. The dependence on $\lambda_{\min}^{(a)}$ in Eq. (6), which approaches a fixed asymptotic value in the $t \to \infty$ limit where posterior vectors $p_t$ are generated from a fixed asymptotic distribution, captures the benefit of more diverse posterior beliefs $p_t^\star$. When observations $x_t \sim p(\cdot|z_t^\star)$ contain little information about the true state $z_t^\star$, posterior beliefs will be more uncertain, decreasing $\lambda_{\min}^{(a)}$, which falls to zero in the limit where posteriors $p_t^\star$ fail to span the space of possible beliefs (e.g. if some latent states are indistinguishable), making $B_{\text{eq}}^{(a)}$ no longer full rank, and hence singular.[7] The $Z$-dependence in Eq. (6) indicates that reward estimation is easier when

---

the latent space is lower dimensional, in which case prior knowledge of the latent structure is more valuable. Lastly, note that the bound probability, Eq. (7), reduces to $1 - \delta$ as $T \to \infty$, but falls to zero at early times.

## 5.2 REGRET BOUND

Since Algorithm 1 uses Thompson sampling with multivariate normal posteriors centered around the estimators $\hat{\mu}^{(a)}$ whose errors are bounded at large $t$ in Theorem 1, we expect that as $t \to \infty$ and these posteriors become sharply peaked, sampled parameters $\mu^{(a)}$ will approach the true values $\mu_\star^{(a)}$, resulting in low regret.

Theorem 2 below demonstrates this, and depends on two important quantities: (1) We define the pairwise reward gap

$$\Delta_{a^\star,a} = ||\mu_\star^{(a^\star)} - \mu_\star^{(a)}||_2 \quad (9)$$

as the Euclidean norm of the difference of mean reward parameters for actions $a^\star$ and $a$. (2) We define the limiting pairwise probability density

$$\rho_{a^\star,a}^{(t)} := \lim_{\epsilon\to 0+}\frac{1}{\epsilon}\cdot\mathbb{P}_{x_{1:t}\sim\rho_{\text{eq}}^{(\phi)}}\Big(p_t^\star \in \mathcal{P}_{a^\star},$$
$$(p_t^\star)^\top(\mu_\star^{(a^\star)} - \mu_\star^{(a)}) < \epsilon||\mu_\star^{(a^\star)} - \mu_\star^{(a)}||_2\Big), \quad (10)$$

which is the probability density that (i) action $a^\star$ is optimal at time $t$, i.e. $a_t^\star := a(p_t^\star) = a^\star$, and (ii) the reward gap between action $a^\star$ and $a$ is infinitesimally small. This quantifies the probability that the sequence of context data $x_{1:t}$ (generated with parameters $\theta^\star, \phi^\star, \mu_\star$) will determine a posterior $p_t^\star$ for which the optimal action is very difficult to resolve. We denote the $t \to \infty$ limit of Eq. (10) – which is well-defined due to the ergodicity and asymptotic equilibration of the latent state – as $\rho_{a^\star,a} := \lim_{t\to\infty}\rho_{a^\star,a}^{(t)}$.

With these definitions, we state our main result:

**Theorem 2.** *Under the same conditions as Theorem 1 (i.e. $\hat{p}_t \to p_t^\star$), and when reward parameter vectors satisfy $||\mu_\star^{(a)}||_1 < u_\mu \in \mathbb{R}^+$ for all $a$, the expected regret incurred by Algorithm 1 (with $\tilde{\sigma}_r = 1$) after $T$ timesteps satisfies the upper bound*

$$\mathcal{R}(T) \leq \frac{8Z^3}{\pi_{\min}^2}\sqrt{\Delta_{\text{likely}}\Delta_{\text{worst}}T} + O(T^{2/5}) \quad (11)$$

*where $\Delta_{\text{worst}} := \max_{a^\star,a}\Delta_{a^\star,a}$ is the worst-case reward gap and*

$$\Delta_{\text{likely}} = 2Z\lambda_{\min}^{-2}\Big(\sigma_{\text{eq}}^2 + \frac{4u_\mu^2}{\gamma_{\phi^\star}}\big(1 + \log\zeta_{\phi^\star}\big)\Big)\sum_{a^\star,a}\frac{\rho_{a^\star,a}}{\Delta_{a^\star,a}}, \quad (12)$$

*with $\rho_{a^\star,a}$ defined above, and other quantities defined in Theorem 1.*

---

will approach the zero matrix, and again $\lambda_{\min}^{(a)} \to 0$ and the bound becomes weak due to less data for action $a$.

---

[6]Recall that $\mathbf{1}(p_t \in \mathcal{P}_a)$ is the binary truth value of the statement that $a = \text{argmax}_{a'}p_t^\top\mu_\star^{(a')}$ is the optimal action given the posterior belief $p_t$.

[7]Furthermore, if an action $a$ is rarely or never optimal, $B_{\text{eq}}^{(a)}$

*Proof (Outline).* Appendix D has the complete proof, and follows several steps: (1) We derive a bound (Lemma D.1) on the probability of a suboptimal action $a_t \neq a_t^\star$ for linear Thompson sampling, under the assumption of an upper bound on the estimation error $|\hat{\mu}^{(a)} - \mu_\star^{(a)}|$. (2) We extend (1) into a high-probability bound on the regret incurred at timestep $t$ (Lemma D.2), by taking an expectation over linear bandit context vectors. (3) We sum over timesteps to bound the cumulative regret (Corollary D.3.1), by decomposing the regret at time $t$ into the "likely" regret $\propto \Delta_{\text{likely}}$ when the per-timestep regret bound holds and a worst-case regret $\Delta_{\text{worst}}$ when it fails with probability $\delta(t)$. We optimize the time-dependent function $\delta(t)$, which reduces regret by a factor of $\sqrt{\Delta_{\text{likely}}/\Delta_{\text{worst}}}$ relative to the worst-case. (4) We use the specific estimator bound from Theorem 1 to apply the linear Thompson sampling regret bound, Corollary D.3.1, to the latent bandit setting, where linear bandit contexts are posterior beliefs, $c_t = p_t^\star$. $\qquad\square$

*Structure of Theorem 2.* The dependence on $\rho_{a^\star,a}/\Delta_{a^\star,a}$ captures the fact that the "likely" regret increases when there is more probability mass for posterior beliefs $p_t^\star$ for which the optimal action is hard to resolve, and that decreasing the reward gap $\Delta_{a^\star,a}$ makes the optimal action still harder to resolve when such posterior beliefs occur. We discuss dependence on the number of actions $K$, as well as the scaling of $\Delta_{\text{likely}}$ with the squared estimation error, Eq. (6), at the end of Appendix D.6.

*Related bandit literature.* Theorem 2 is a bound for linear Thompson sampling, applied to the case where the context vectors are posterior probability vectors in a latent bandit problem, $c_t = p_t^\star$ (see Lemma 1). While this limits its applicability in the latent bandit setting to the case where $(\theta^\star, \phi^\star)$ are known, the bound is novel in relation to existing regret bounds, in three ways:

- *Problem-dependence.* Eq. (12) describes the influence of task parameters $(\theta^\star, \phi^\star, \mu_\star)$ on regret, via their influence on posterior beliefs $p_t^\star$ (in $\rho_{a^\star,a}$), and resulting reward uncertainty ($\lambda_{\min}^{-1}$).

- *Heavy-tailed reward distributions.* Theorem 2 makes no assumptions (e.g. sub-Gaussianity) about the reward distribution. This is because the derivation of Theorem 1 relies mainly on Chebyshev's inequality – which only assumes a finite variance for reward distributions – to bound the estimator error and covariance.[8]

- *Non-stationarity.* Our result applies in a non-stationary linear bandit setting where contexts $c_t = p_t^\star$ are correlated across time (via latent state dynamics).[9]

Our results compare to notable existing works as follows:

- *Linear Thompson sampling.* The problem-dependent regret bound for linear Thompson sampling [Agrawal and Goyal, 2013a] is $O(\log T)$, but this and most subsequent works assume sub-Gaussian (and i.i.d.) rewards.

- *Heavy-tailed reward distributions.* Some works [Medina and Yang, 2016, Xue et al., 2021] have obtained problem-independent $O(T^{1/2+\epsilon})$ regret bounds with heavy-tailed rewards. In comparison, our bound captures problem-dependent structure in a more general setting with non-stationarity and latent variables.

- *Non-stationary bandits.* Existing bounds [Luo et al., 2018, Hong et al., 2020] depend on the number of change-points; when distribution changes occur at a constant rate (e.g. due to latent state changes), these bounds are $\Omega(\sqrt{T})$ or linear, in contrast to our $O(\sqrt{T})$ bound.

- *Latent bandits.* The Thompson sampling regret bounds of Hong et al. [2020] are complementary to ours, in that (i) they are problem-independent and assume sub-Gaussian rewards, but more importantly (ii) they assume an alternative definition of regret, relative to an oracle policy which sees the true latent state. (Note that, in contrast to our $O(\sqrt{T})$ regret, regret relative to such an oracle cannot be sublinear as long as latent state changes occur at a constant rate.)

Extending our regret bound to the case where $(\theta^\star, \phi^\star)$ are learned would be straightforward[10] given a bound on the posterior error $|\hat{p}_t - p_t^\star|$. (We are not aware of such convergence guarantees for online EM applied to HMMs.)

# 6 EXPERIMENTS

In order to demonstrate the strong performance of our algorithms, we conduct experiments to compare the L$^2$TS and L$^2$UCB algorithms[11] with relevant baselines on (i) discrete latent bandit tasks with synthetic data, and (ii) a Gaussian latent bandit problem for a mining application involving real data. In all cases, the true initial state distribution $p_0^\star(z)$ differs at random from the model initial state distribution $p_0(z)$ (see Appendix B).

**Multinomial Context and Reward Distributions.** *Problem 1.* In this problem, $Z = 2$, $K = 2$, and $x_t \in \{1, ..., X\}$ with $X = 4$, and with $\Phi^\star = \left( \begin{smallmatrix} 0.9 & 0.1 \\ 0.1 & 0.9 \end{smallmatrix} \right)$. We used 5 offline samples $x \sim p(x|z)$ for each $z$ to improve the initial estimate $\hat{\theta}$ at $t = 0$ for both L$^2$TS and L$^2$UCB. *Problem 2.* In

---

[8]In the case of sub-Gaussian rewards, we expect the Theorem 1 to hold with higher probability than Eq. (7), as outlier rewards are exponentially rare. This will improve the $O(\sqrt{T})$ scaling of Theorem 2.

[9]The limiting case where contexts $c_t$ are i.i.d. can effectively be obtained by making the mixing rate large, $\gamma_{\phi^\star} \to \infty$, in which case intrinsic reward noise dominates, $\mathcal{R}(T) \propto \sigma_{\text{eq}}$.

[10]Lemmas D.2 and D.3 bound the regret of linear Thompson sampling in the case where the agent's contexts $\hat{c}_t = \hat{p}_t$ deviate from the true contexts $c_t = p_t^\star$ which determine expected rewards.

[11]Code will be made available at github.com/elliotnelson/hmm-bandits.

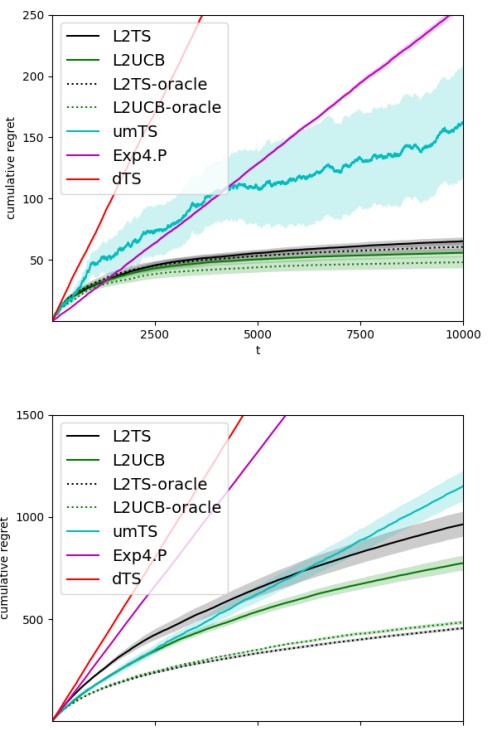

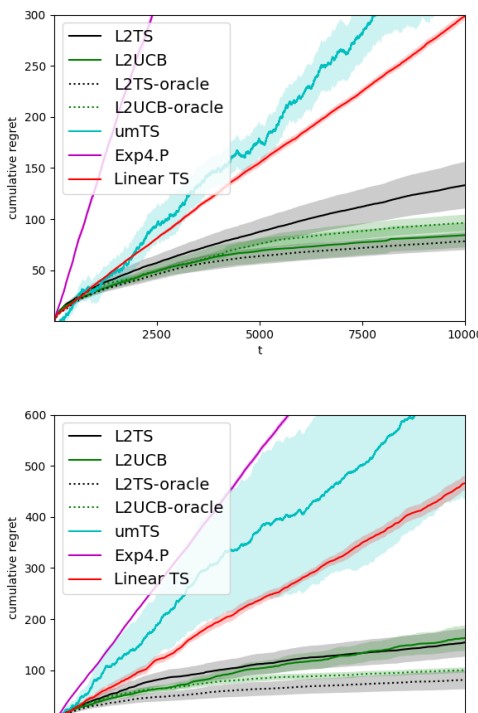

Figure 2: **Top:** Mean cumulative regret for a synthetic task with discrete categorical variables (Problem 1). Shaded regions show uncertainty in the mean over 10 episodes. **Bottom:** Results for a synthetic task with clustered contexts (Problem 2).

Figure 3: **Top:** Mean cumulative regret in a Gaussian-variable rover mining task. Shaded regions show uncertainty in the mean over 10 episodes. **Bottom:** As above but with a rarely changing latent state (nearly diagonal $\Phi^\star$).

this problem, $(Z, X, K) = (4, 12, 8)$, with Bernoulli reward probabilities sampled uniformly in $(0, 1)$, $\phi^\star_{z,z} = 0.75$ on-diagonal and uniform off-diagonal, and contexts clustered into groups which are only emitted by a single latent state. (See Appendix B.1 for more details to both problems.)

**Mining Application.** We consider an application where a rover explores and mines for oxide ore. The rover travels over various blocks of land taking x-ray fluorescent meter samples (context $x$), which provide information about the oxide grade, which in turn depends on the presence of one of three latent geological classes (latent state $z$). Non-stationarity in this application is from spatial dependence between adjacent blocks of land. We assume the rover chooses between two mining strategies for different minerals (actions $a$), such that there are varying reward probabilities depending on uncertain revenue from the mined ore as well as fixed and variable costs. We provide numerical details about the latent bandit model parameters in Appendix B.2, highlighting in particular how the context distribution $p(x|z)$ is obtained using real-world geological data [Eidsvik et al., 2015].

**Baselines.** We compare L$^2$TS and L$^2$UCB with three baselines (see Appendix B for all parameter settings): (1) Uncertain Model Thompson Sampling (umTS): We adapt Algorithm 3 of Hong et al. [2020] – which uses particle filtering to maintain a posterior over reward models, latent states, and latent transition matrices – to our setting by using oracle knowledge of $p(x_t|z; \theta^\star)$ for additional posterior updates, which we denote in Figures 2 and 3 with the label umTS$^\star$. (In the graphical setting of Hong et al. [2020], the latent state only influences rewards, and not contexts.) (2) Exp4.P [Beygelzimer et al., 2010]: We use expert advisor classifiers trained (with varying latent state distributions) to label contexts $x$ according to corresponding optimal actions, as detailed in Appendix B, and modify the weight update of Exp4.P to discount the influence of old context data on current weights assigned to experts, and use the true dynamics timescale to set the discount factor. (3) Discounted Thompson Sampling (dTS) [Raj and Kalyani, 2017]: We extend dTS to maintain success ($r = 1$) and failure ($r = 0$) counts for each discrete context-action pair $(x, a)$, and (like Exp4.P) allow dTS to use the true dynamics timescale to set the discount factor $\gamma$. (We only include dTS in the experiment with discrete context variables.)

We also compare to oracle variants of L$^2$TS and L$^2$UCB which use the true posterior $p_t^\star$ (i.e. condition on the true parameters $\theta^\star, \phi^\star, \mu_\star$) instead of the estimate $\hat{p}_t$. As such, the oracle variants are simply linear Thompson sampling and LinUCB with uncorrupted or unbiased vectors $c_t = p_t^\star$. (For this reason, the L$^2$TS oracle satisfies the conditions for Theorems 1 and 2.) Lastly, in the rover mining experiment, we also compare to linear Thompson sampling using the raw contexts $x_t$ (instead of posteriors $\hat{p}_t$ or $p_t^\star$).

**Results.** Figures 2 and 3 show the cumulative regret for all algorithms, averaged over 10 episodes, for (respectively) the categorical-variable synthetic tasks and the Gaussian-variable rover mining tasks. L$^2$TS significantly outperforms baselines. While umTS models the true latent structure and is given additional prior knowledge of $\theta^\star$, it struggles relative to our algorithms except in the low-dimensional task (Problem 1), possibly due to challenges of scaling particle filtering to higher dimensions. Exp4.P suffers from asymptotically linear regret due to its inability to model the underlying latent dynamics.[12] Discounted TS performs most poorly in Figure 2 due to its inability to model the latent space or to transfer information gained across different discrete contexts. The poor performance of linear Thompson sampling relative to L$^2$TS in Figure 3 shows the benefit of using the (history-dependent) posterior probabilities $p_t$ as contexts for linear reward estimation, instead of the directly observed contexts $x_t$. In most cases, the asymptotic performance of L$^2$TS and L$^2$UCB is comparable to their respective oracle variants (differing mainly in the overhead cost incurred at early times), indicating that approximation error in the learned transition probabilities and context distributions is under control. (See Appendix B.2 for additional results on parameter estimation error.)

## 7 CONCLUSION & DISCUSSION

In this paper, we have developed a novel multi-armed bandit algorithm for environments with a dynamical latent state influencing both observations (contexts) and rewards. Our algorithm uses prior knowledge of latent graphical structure to transform a nonlinear and non-stationary contextual bandit problem into a linear bandit problem, exploiting the linearity between rewards and posterior probabilities over the latent state. While we considered a specific method (Online EM) to learn the latent transition matrix and context distributions, with specific linear bandit algorithms (LinTS, LinUCB), the high-level approach of treating a posterior belief over latent variables (or over unknown parameters) as context information is general; it can be applied with any method for sequential Bayesian inference, and with other

---

[12]While Exp4.P can leverage statistical correlation between contexts and rewards that is modeled by its expert advisors, it cannot learn the temporal structure of this correlation, which is governed by the latent state.

sequential decision-making algorithms. Our theoretical analysis underscores the influence of the latent dynamics and distributional structure of the environment on task difficulty. Directions for future work include online learning of the latent space dimensionality, application of HMM learning convergence guarantees [Hsu et al., 2012] to non-stationary bandit problems, and extensions of our methodology to partially observable Markov decision process (POMDP) settings or to more complex graphical models.

### Acknowledgements

We are grateful to Karthikeyan Shanmugam for conversations regarding regret analysis and algorithm development.

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
