# OpenReview forum: "Linearizing Contextual Bandits with Latent State Dynamics"
_auai.org/UAI/2022/Conference — UAI 2022 Poster_

### Official Review · Reviewer_bT3S · 2022-03-31

**Q2(1) Originality/Novelty:** 2
**Q2(2) Significance/Impact:** 2
**Q2(3) Correctness/Technical Quality:** 3
**Q2(6) Clarity Of Writing:** 3
**Q6 Overall Score:** 6
**Q8 Confidence In Your Score:** 3

**Q1 Summary And Contributions:**

The paper studies non-stationary MAB where the reward is determined by a latent state, which has a latent dynamic that can be inferred from the contexts&rewards. Focusing on finite states&arms, the reward is linear in the indicator vector of the state-action pair and hence its posterior vector. The algorithm then combines two existing methods: online EM & linear-UCB/TS. Regret bounds are proved when two unknown parameters are known exactly. Simulation shows improved performance over baselines.

**Q10 Ethical Concerns (Optional):**

I do not have any major concerns.


**Q2 Assessment Of The Paper:**

More detailed information regarding each of these aspects is given below:

**Q2(4) Quality Of Experiments (Optional):**

2: Fair: The experimental evaluation is weak: important baselines are missing, or the results do not adequately support the main claims.

**Q2(5) Reproducibility:**

2: Fair: Key resources (e.g., proofs, code, data) are unavailable but key details (e.g., proof sketches, experimental setup) are sufficiently well-described for an expert to confidently reproduce the main results.

**Q3 Main Strengths:**

1. The combination of HMM and MAB is novel, and I think overall the intersection between causal models and bandits is a promising direction.
2. The proposed method and theory is overall sound.
3. The presentation is clear.


**Q4 Main Weakness:**

1. The practical impact is unclear to me, without a strong motivating example / real data-based experiments.
3. The absence of code may affect the reproducibility.
4. The absence of several baselines affect the quality of the numerical experiments: (i) as the main arguments in the title/introduction is that the context may have a nonlinear relationship with the reward, related baselines (e.g., GP-TS, Neural TS) would make the statements much more stronger. linear-TS would also be a useful baseline. (ii) a variant that does not know the latent state but know the two sets of unknown parameters would be another useful baseline that is consistent with your theory.


**Q5 Detailed Comments To The Authors:**

Besides my comments in Q4 which I hope the authors can address, I have several minor questions/comments/suggestions:


Questions/suggestions:

1. Why only prove the theory for TS?
2. The paper critically relies on the finite-state assumption. Anyway we can extend to, e.g., continuous space?
3. When we talk about the approximate posterior inference in bandits, the approximation error is always a concern and whether it will cause the algorithm to escape from the local optimum. I hope to see some discussions and clarifications.
4. The theory does not require ANY assumption on the reward distributions is quite surprising. I hope to see some discussions on why we can achieve that.
5. Discussions on the assumption `prior over latent state is known`. Is it critical? Will it be less important when T grows? I would like to see it as a baseline as well.
6. Regarding a naive application of contextual bandits, I think the non-statioanrity induced by the confounders would be more essential than the nonlinearity. Please correct me if I am wrong. Otherwise, I suggest to add the discussion.


Typos

1. `[REFS]` in Section 2
2. the notations after `We collectively denote the action-wise` in Section 3.1
3. `apply linear Thompson Sampling [3]. Upper-Confidence Bounds` in Section 4
1. POMDP in the discussion section is not defined (although commonly known).

**Q7 Justification For Your Score:**

Overall, I think the paper is a decent contribution to the bandit literature and I appreciate the combination of the causal inference area and the bandit problems. As the contribution is incremental (a combination of existing methods) in a specific model, and the practical relevance is unclear to me, I think it should be evaluated as having `moderate impact` and hence I give 6.


**Q9 Complying With Reviewing Instructions:**

1: Yes.

---

### Official Review · Reviewer_HiDx · 2022-04-09

**Q2(1) Originality/Novelty:** 3
**Q2(2) Significance/Impact:** 2
**Q2(3) Correctness/Technical Quality:** 3
**Q2(6) Clarity Of Writing:** 3
**Q6 Overall Score:** 5
**Q8 Confidence In Your Score:** 2

**Q1 Summary And Contributions:**

In this paper, an approach is proposed to reduce the contextual multi-armed bandit problems with an evolving hidden state to a linear bandit problem under some conditions, which then is used to solve the bandit tasks. Finally, experiments were conducted on two simple bandit problems to verify the effectiveness of the proposed approach.

**Q2 Assessment Of The Paper:**

More detailed information regarding each of these aspects is given below:

**Q2(4) Quality Of Experiments (Optional):**

2: Fair: The experimental evaluation is weak: important baselines are missing, or the results do not adequately support the main claims.

**Q2(5) Reproducibility:**

3: Good: Key resources (e.g., proofs, code, data) are available and key details (e.g., proofs, experimental setup) are sufficiently well-described for competent researchers to confidently reproduce the main results.

**Q3 Main Strengths:**

The paper is well-motivated and the proposed approach also sounds technically reasonable.

**Q4 Main Weakness:**

1. The proposed approach seems to rely heavily on accurate estimations of the latent state transition matrix and an observation model. In the paper, such models were learned via online EM algorithms, it is unclear whether the accuracy of the learned models can be guaranteed.

2. The proposed approach is only verified on two very simple toy environments, the scalability of the proposed framework is unclear.

3. In my view, some POMDP-related methods can also be applied to the non-linear-bandit problems mentioned in the paper, which have successful applications in some large-scale domains, then what are the advantages of the proposed framework compared with some POMDP-related techniques?


**Q5 Detailed Comments To The Authors:**

In this paper, a multi-armed bandit algorithm for environments with a dynamical latent state has been proposed, where prior knowledge of latent graphical structure is used to transform a nonlinear contextual bandit problem into a linear bandit problem, and the linearity between rewards and posterior probabilities over the latent state is exploited.

In general, the paper is well-motivated and sounds technically reasonable. However, as mentioned in the response to Q4, the effectiveness and correctness of the proposed approach rely heavily on some strong assumptions, e.g., the accurate estimations of the latent state transition matrix and an observation model. Moreover, the proposed approach is only verified on two very simple toy environments, and it is unclear the advantages of the proposed framework compared with some POMDP-related techniques, please see the details in the response to Q4. Also, it seems that the relationship between the performance of the proposed technique and the accuracy of the learned matrix and model should also be discussed and presented.


-----------------------Post Rebuttal-------------------------

After reading the authors' responses and the other reviewers' comments,  the authors did address some of my concerns, and my rating is changed to borderline accept.

**Q7 Justification For Your Score:**

In general, the paper is well-motivated and sounds technically reasonable. However, as mentioned, the effectiveness and correctness of the proposed approach rely heavily on some strong assumptions. Moreover, the proposed approach is only verified on two very simple toy environments, and it is unclear the advantages of the proposed framework compared with some POMDP-related techniques.


**Q9 Complying With Reviewing Instructions:**

1: Yes.

---

### Official Review · Reviewer_AHbZ · 2022-04-12

**Q2(1) Originality/Novelty:** 2
**Q2(2) Significance/Impact:** 2
**Q2(3) Correctness/Technical Quality:** 2
**Q2(6) Clarity Of Writing:** 1
**Q6 Overall Score:** 5
**Q8 Confidence In Your Score:** 3

**Q1 Summary And Contributions:**

To handle the non-stationarity problem from confounding latent variables, the authors reduced the latent contextual bandit problem to the linear bandit setting using the prior knowledge of causal structure. Then, combining the online EM method for HMM with the linear bandit method, they proposed two algorithms, Latent Linear Thompson Sampling (L2TS) and Latent Linear UCB (L2UCB). They gave theory analysis, including the error upper bound or the regret bound.

**Q2 Assessment Of The Paper:**

More detailed information regarding each of these aspects is given below:

**Q2(4) Quality Of Experiments (Optional):**

2: Fair: The experimental evaluation is weak: important baselines are missing, or the results do not adequately support the main claims.

**Q2(5) Reproducibility:**

2: Fair: Key resources (e.g., proofs, code, data) are unavailable but key details (e.g., proof sketches, experimental setup) are sufficiently well-described for an expert to confidently reproduce the main results.

**Q3 Main Strengths:**

The authors focused on the contextual multi-armed bandit problems with the presence of a latent variable, which is crucial and challenging in many domains. They used prior knowledge about the graphical model to reduce the problem into a linear bandit one. They then proposed two algorithms with theoretical analysis, a high-probability error bound and a problem-dependent regret bound. The proofs look detailed and complete.

**Q4 Main Weakness:**

The condition under which contextual multi-armed bandit problems with an evolving hidden state can be mapped to a linear bandit problem is strong and hard to satisfy in real-world applications. How could we determine whether this condition holds or not in applications? If such a condition fails to hold, the proposed method may deteriorate much. It might be better to test the performance in simulations.

Some notations in the paper are confusing, which makes the readers hard to follow the paper. Please see Q5. And the code is unavailable.

**Q5 Detailed Comments To The Authors:**

- Some notations are a bit confusing. For example, $\mu_*^{(a)} \in \mathbb{R}^Z$ in Section 3.1 but in Section 3.2, $\mu_*^{(a)} \in \mathbb{R}^d$.

- There is no description about ${\bf{1}}(a_{t'}=a)$ below Eq.(4). Is it an all-ones matrix? In Eq.(4), what if the covariance matrix $B^{(a)}$ is not reversible?

- In Page 3, at the bottom of the right column, $\mu(a) \thicksim \mathcal{N} (\hat{\mu}(a),(B(a))^{−1}$ lacks a right parenthesis.

- “but need satisfy no other conditions”---> “but need to satisfy no other conditions”.

- In the equation below Eq.(5), $(\mu_*^{(a)})z$ needs to be $(\mu_*^{(a)})_z$, where $z$ should be subscript?

- What is the relationship between the observations/contexts $x$ and $c$? I suppose they are the same when I first read Section 3.1, but it seems not true afterward. In Section 3.2, $c_t$ is a context feature vector; in Section 3.3, $c_t$ is defined as a vector with elements equal to the posterior probabilities; in Section 5.1, $c \approx 6.8$. These confusing notation examples really hinder the readability of this paper.



**Q7 Justification For Your Score:**

Please see Q3-Q5. Though there are many minor mistakes or typos in the paper, it can be seen that this paper has made some key contributions. So I choose borderline accept.

**Q9 Complying With Reviewing Instructions:**

1: Yes.

---

### Decision · Program_Chairs · 2022-05-15

**Decision:**

Accept (Poster)

**Comment:**

Meta Review: The reviewers in general, find the paper interesting. There are questions regarding the strictness of the assumptions and scalability of the method. However, the author answered major questions. And EM is not the focus but a reasonable choice here where the scalability is a more general question. I thus recommend weak accept.